# Retinoic acid-gated BDNF synthesis in neuronal dendrites drives presynaptic homeostatic plasticity

**Shruti Thapliyal†, Kristin L Arendt†, Anthony G Lau, Lu Chen***

Departments of Neurosurgery, Neuropsychiatry and Behavioral Sciences, Stanford University School of Medicine, Stanford, United States

**Abstract** Homeostatic synaptic plasticity is a non-Hebbian synaptic mechanism that adjusts synaptic strength to maintain network stability while achieving optimal information processing. Among the molecular mediators shown to regulate this form of plasticity, synaptic signaling through retinoic acid (RA) and its receptor, RARα, has been shown to be critically involved in the homeostatic adjustment of synaptic transmission in both hippocampus and sensory cortices. In this study, we explore the molecular mechanism through which postsynaptic RA and RARα regulates presynaptic neurotransmitter release during prolonged synaptic inactivity at mouse glutamatertic synapses. We show that RARα binds to a subset of dendritically sorted brain-derived neurotrophic factor (*Bdnf*) mRNA splice isoforms and represses their translation. The RA-mediated translational de-repression of postsynaptic BDNF results in the retrograde activation of presynaptic tropomyosin receptor kinase B (TrkB) receptors, facilitating presynaptic homeostatic compensation through enhanced presynaptic release. Together, our study illustrates an RA-mediated retrograde synaptic signaling pathway through which postsynaptic protein synthesis during synaptic inactivity drives compensatory changes at the presynaptic site.

## Editor's evaluation

This study defines an important homeostatic pathway at excitatory synapses where retinoic acid-dependent signaling in response to synaptic inactivity drives the local synthesis of the neurotrophin BDNF in dendrites, which in turn is released to adaptively modify neurotransmitter release properties at presynaptic terminals. The authors use genetic tools to localize the action of distinct components of the pathway to pre- vs post-synaptic compartments and use biochemical approaches to define a molecular link between RA and the local translation of specific BDNF transcripts. The experiments have been well-executed and the compelling findings fill a gap in our knowledge about how presynaptic function is adaptively modulated by retrograde BDNF signaling by highlighting the role of RA in this process.

## Introduction

Activity in neural circuits is highly dynamic and is continuously subjected to experience-dependent changes due to synaptic plasticity. Information processing that optimally interprets the external world is vital for the survival of an organism and requires precisely timed and well-orchestrated excitatory and inhibitory synaptic transmission in neural circuits. Hebbian plasticity is considered to be the cellular mechanism underwriting memory formation. By contrast, homeostatic synaptic plasticity counters the self-reinforcing nature of Hebbian plasticity, thus maintaining network stability while preserving its capacity for information processing. Retinoic acid (RA), a well-documented developmental morphogen,

**\*For correspondence:**
luchen1@stanford.edu

†These authors contributed equally to this work

has emerged in recent studies as a critical molecular component in homeostatic synaptic plasticity (*Chen et al., 2014*). Chronic synaptic inactivity at glutamatergic synapses triggers RA synthesis in neuronal dendrites, which in turn drives local synthesis and synaptic incorporation of GluA1 containing α-amino-3-hydroxy-5-methyl-4-isoxazolepropionic acid receptors (AMPARs) and concomitant removal of γ-aminobutyric acid receptors (GABA$_A$Rs) from inhibitory synapses, thus restoring synaptic excitatory/inhibitory balance (*Aoto et al., 2008*; *Sarti et al., 2013*; *Maghsoodi et al., 2008*). The RA receptor RARα is a nuclear receptor that mediates RA-dependent transcriptional activation during development (*Chambon, 1996*). In mature neurons, however, RARα translocates out of the nucleus and acts as a molecular mediator for RA-dependent de novo protein synthesis (*Poon and Chen, 2008*). In the context of homeostatic synaptic plasticity, RARα has been shown to bind to mRNAs via specific sequence motifs, and mediate the translation of these mRNAs in an RA-dependent manner. For example, activation of GluA1 synthesis upon chronic synaptic activity blockade has been demonstrated to require both RA synthesis and normal RARα expression (*Aoto et al., 2008*; *Maghsoodi et al., 2008*; *Wang et al., 2011*).

Several aspects of RA-mediated homeostatic regulation of postsynaptic function have been addressed in previous studies (*Sarti et al., 2013*; *Maghsoodi et al., 2008*; *Poon and Chen, 2008*). Additionally, changes in presynaptic functions (i.e. release probability) as part of homeostatic synaptic mechanism have also been documented in multiple organisms (*Wang et al., 2011*; *Lindskog et al., 2010*; *Thiagarajan et al., 2005*; *Davis and Müller, 2015*; *Murthy et al., 2001*; *Gong et al., 2007*; *Jakawich et al., 2010*). Compared to *Drosophila* neuromuscular junctions (a type of glutamatergic synapse exhibiting robust presynaptic homeostatic plasticity), less is known about the signaling pathways involved in the homeostatic presynaptic changes at mammalian synapses. One of the classical synaptic retrograde messengers, brain-derived neurotrophic factor (BDNF), was found to be involved in the homeostatic regulation of presynaptic release in cultured hippocampal neurons (*Jakawich et al., 2010*). Moreover, both postsynaptic AMPAR blockade-induced activation of RA signaling and the phospholipase D (PLD)-mTORC1 signaling pathway have been implicated in dendritic protein synthesis in the context of homeostatic plasticity (*Henry et al., 2012*; *Henry et al., 2018*).

In this study, we explore whether and how synaptic RA/RARα signaling modulates postsynaptic BDNF synthesis during chronic synaptic inactivity in hippocampal pyramidal neurons. We show that RARα is directly associated with specific dendritically localized *Bdnf* mRNA isoforms. This enables RA-induced translational de-repression of BDNF synthesis in dendrites, resulting in retrograde activation of presynaptic tropomyosin receptor kinase B (TrkB) receptors and upregulation of miniature excitatory postsynaptic current (mEPSC) frequencies. Using pre- and postsynaptic specific genetic deletions of *Rara*, *Bdnf*, and *Ntrk2*, we unequivocally established the locations of the actions of each of these key signaling molecules during homeostatic modulation of presynaptic function.

## Results

### Postsynaptic RA receptor RARα mediates activity blockade-induced presynaptic homeostatic plasticity

While homeostatic plasticity studies conducted in neuronal cultures consistently report an increase in mEPSC amplitude in response to chronic synaptic silencing, the increase in mEPSC frequency has only been observed in those using older and likely more mature cultures (*Aoto et al., 2008*; *Wang et al., 2011*; *Jakawich et al., 2010*; *Turrigiano et al., 1998*). We have previously shown that in older (21 days in vitro [DIV]) but not younger (14 DIV) cultured primary hippocampal neurons, chronic treatment with postsynaptic activity blockers such as the AMPAR antagonist 6-cyano-7-nitroquinoxaline-2,3-dione (CNQX) or L-type voltage-dependent calcium channel (VDCC) inhibitor nifedipine results in a significant increase in both mEPSC frequency and mEPSC amplitude (*Wang et al., 2011*). Thus, consistent with the literature, while the increase in mEPSC amplitude is observed in both young and old cultured neurons, the enhancement in frequency is only present in the older cultures (*Wang et al., 2011*). Importantly, this increase in both mEPSC frequency and amplitude requires RA synthesis as pharmacological inhibition of RA synthesis blocks the increase in both parameters. Additionally, RA synthesis in postsynaptic neurons triggers presynaptic changes in a cell-autonomous manner as sparse expression of a dihydropyridine-insensitive L-VDCC, which specifically blocks RA synthesis in transfected postsynaptic neurons, prevented the nifedipine-induced homeostatic increase in mEPSC frequency. However,

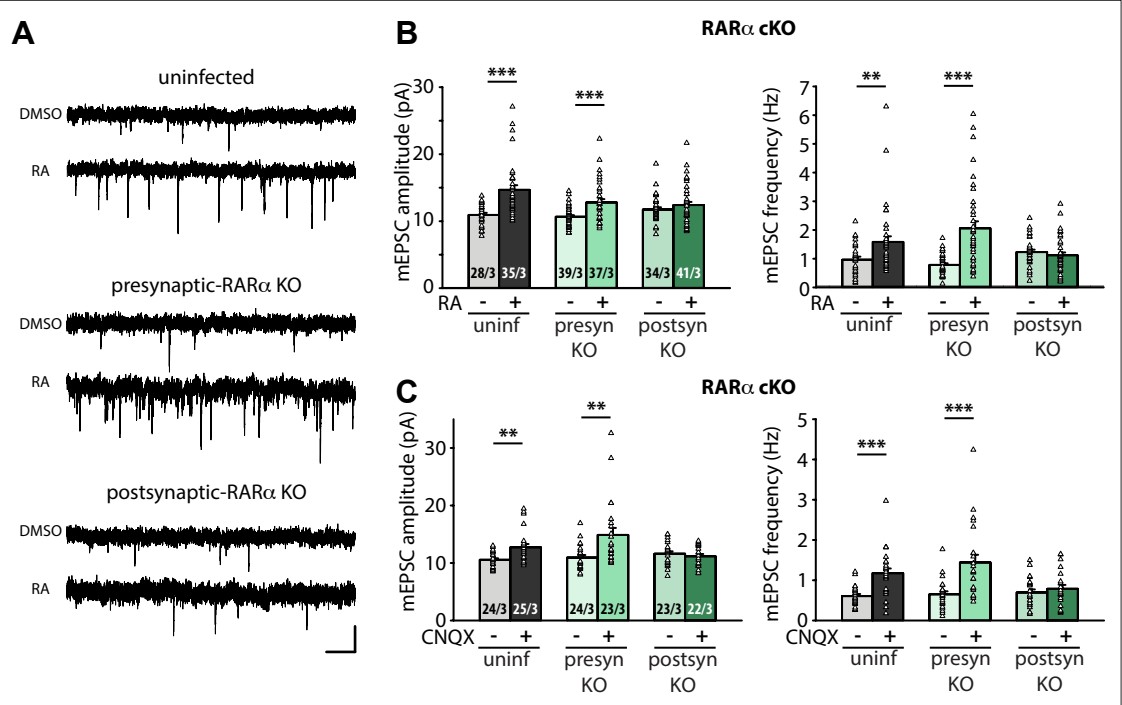

**Figure 1.** Postsynaptic RARα expression is required for presynaptic homeostatic plasticity. (**A**) Example traces of mEPSCs recorded from hippocampal pyramidal neurons in organotypic slices from WT (uninfected), presynaptic RARα KO (Cre expression in CA3) and postsynaptic RARα KO (Cre expression in CA1) groups treated with DMSO or RA (10 μM, 4 hr). Scale bars, 10 pA, 0.5 sec. (**B**) Quantification of mEPSC amplitudes and frequencies recorded from WT, presynaptic and postsynaptic RARα KO neurons treated with DMSO or RA. **, $p < 0.01$; ***, $p < 0.001$; two-way ANOVA followed by Mann Whitney test. Amp: $F_{(2,208)} = 5.413$, $p < 0.01$; freq: $F_{(2,208)} = 11.81$, $p < 0.0001$. (**C**) Quantification of mEPSC amplitudes and frequencies recorded from WT, presynaptic and postsynaptic RARα KO neurons treated with DMSO or CNQX (36 hours). **, $p < 0.01$; ***, $p < 0.001$; two-way ANOVA followed by Mann Whitney test. Amp: $F_{(2,135)} = 6.004$, $p < 0.01$; freq: $F_{(2,135)} = 5.23$, $p < 0.01$. n/N represent number of neurons/number of independent experiments (pups). All graphs represent mean ± SEM.

The online version of this article includes the following source data and figure supplement(s) for figure 1:

**Source data 1.** Individual data spreadsheet in *Figure 1B and C*.

**Figure supplement 1.** Additional data related to *Figure 1*: paired-pulse ratio and failure rate of evoked excitatory postsynaptic currents (eEPSCs) at CA3-CA1 synapses in retinoic acid (RA)-treated slices; viral expression efficacy in CA3; homeostatic plasticity of mIPSCs in older neurons.

**Figure supplement 1—source data 1.** Individual data spreadsheet in Fig.S1A, S1B and S1D.

what remains unanswered is where and through what molecular signaling pathway RA acts to trigger presynaptic changes.

To first establish that RA also increases mEPSC frequency in older organotypic cultured hippocampal slices, we measured mEPSCs in WT neurons from 21–25 DIV slice cultures after acute RA treatment (10 μM, 4 hr). A significant increase in both mEPSC amplitude and frequency in CA1 pyramidal neurons was observed (*Figure 1A and B*), replicating our previous findings in primary hippocampal neuronal cultures (*Wang et al., 2011*). To further understand the cellular mechanisms driving mEPSC frequency changes, we measured paired-pulse ratios (PPRs) of the evoked EPSCs (eEPSCs). RA treatment significantly reduced the PPRs, suggesting potential enhancement of presynaptic function by RA (*Figure 1—figure supplement 1A*). We have previously reported the induction of silent synapses during tetrodotoxin (TTX)-induced homeostatic synaptic plasticity (*Arendt et al., 2013*). To understand whether activation of silent synapses may be contributing to the mEPSC frequency increases by RA, we measured eEPSC failure rate at both –60 and +40 mV using a minimal stimulation protocol (*Arendt et al., 2013*). The failure rates were comparable between –60 and +40 mV in both DMSO- and RA-treated slices, indicating little if any presence of postsynaptic silent synapses in these slices, and that activation of postsynaptic silent synapses is not a contributing factor to the mEPSC frequency changes observed (*Figure 1—figure supplement 1B*). Moreover, given that the mEPSC frequency increase only occurs in older slices, but not in younger ones where synaptogenesis is more robust,

we think it is unlikely that new synapse formation underlies frequency increase. Taken together, these results are consistent with previous findings from both rodent and *Drosophila* studies (*Davis and Müller, 2015*; *Jakawich et al., 2010*; *Subramanian and Dickman, 2015*; *Bergquist et al., 2010*) and demonstrate that modification of presynaptic release is a conserved homeostatic mechanism operating at glutamatergic synapses across species.

We next sought to establish the location of RA's action using organotypic hippocampal slices from RARα conditional knockout mouse (*Chapellier et al., 2002*; *Sarti et al., 2012*). We injected Cre-expressing AAVs into either the CA3 or CA1 regions of the hippocampus to achieve pre- or postsynaptic-specific deletion of RARα in the Schaeffer collateral-CA1 synapses. Care was taken to achieve high infection efficiency in CA3 regions to ensure appropriate interpretation of results from presynaptic Cre-dependent deletion (*Figure 1—figure supplement 1C*). Selective deletion of RARα in postsynaptic neurons (Cre expressed in CA1 neurons), but not presynaptic neurons (Cre expressed in CA3 neurons), blocked RA-induced homeostatic increases in both mEPSC amplitude and frequency (*Figure 1A-B*). Importantly, these results were also observed when homeostatic plasticity was induced with chronic synaptic activity blocker CNQX (20 µM, 36 hr) (*Figure 1C*), which has been shown to induce de novo RA synthesis (*Wang et al., 2011*). Thus, newly synthesized RA acts via postsynaptic RARα to promote homeostatic adjustment of synaptic strength in both pre- and postsynaptic compartments.

RARα signaling has also been shown to mediate homeostatic plasticity at inhibitory synapses (*Sarti et al., 2013*). We thus asked whether the presynaptic enhancement of synaptic strength also occurs at inhibitory synapses. Acute RA treatment in 21–25 DIV cultured slices significantly reduced mIPSC amplitudes as has been shown in younger slices (*Sarti et al., 2013*; *Figure 1—figure supplement 1D*). Interestingly, unlike those in younger neurons, the mIPSC frequency in these older neurons was also reduced by RA. These changes in inhibitory synaptic transmission were blocked by postsynaptic RARα deletion (*Figure 1—figure supplement 1D*). This result is consistent with our previous observations showing that visual deprivation reduces both mIPSC amplitude and frequency in post-critical period layer 2/3 visual cortical neurons (*Zhong et al., 2018*), indicating that the impact of RA on inhibitory synaptic transmission may be conserved across different brain regions. Given the differences in the nature of RA's action on the presynaptic function of excitatory and inhibitory synapses, we focused the rest of the study on molecular mechanisms by which RA drives the enhancement of presynaptic function at excitatory synapses in the hippocampal neurons.

## RA activates BDNF synthesis through direct association between RARα and specific splice isoforms of dendritically localized *Bdnf* mRNAs

A postsynaptically initiated mechanism of translational regulation mediated by RA-RARα signaling modulating presynaptic neurotransmitter release suggests the existence of a retrograde messenger molecule. BDNF is one of the most widely expressed and well-characterized neurotrophins in the developing and adult mammalian central nervous system (*Hofer et al., 1990*; *Conner et al., 1997*). It is synthesized as a proneurotrophin known as proBDNF and secreted as a mixture of proBDNF and mature BDNF processed from proBDNF (*Foltran and Diaz, 2016*). The retrograde BDNF-TrkB signaling has been well established in modulating both Hebbian synaptic plasticity (*Andero et al., 2014*; *Guo et al., 2018*) and homeostatic synaptic plasticity (*Jakawich et al., 2010*). BDNF expression in the brain is developmentally regulated and exhibits a steep increase between postnatal day 15 and 20 (*Schoups et al., 1995*; *Dincheva et al., 2016*), a time period that coincides with the emergence of RA-dependent presynaptic homeostatic modulation observed in 21 DIV primary cultured hippocampal neurons (*Wang et al., 2011*). Immunoblotting of proBDNF in hippocampal tissue collected at different developmental stages showed a gradual increase in BDNF expression during the first 4 postnatal weeks (*Figure 2—figure supplement 1A*). Immunoblot analysis from cultured hippocampal slices (prepared from postnatal day 8 mouse pups) showed a similar trend where the expression of BDNF gradually increases as the cultures mature from 1 to 21 DIV (*Figure 2A*), suggesting that this developmental upregulation of BDNF expression is preserved in our organotypic hippocampal slice culture system.

We next sought to investigate whether postsynaptic RA/RARα is involved in retrograde BDNF signaling through regulation of BDNF synthesis. We first explored the possibility that RARα directly binds to *Bdnf* transcripts. In rodents, the *Bdnf* gene represents a complex structure that consists of nine unique promoters (on nine separate exons) that drive the expression of several distinct *Bdnf*

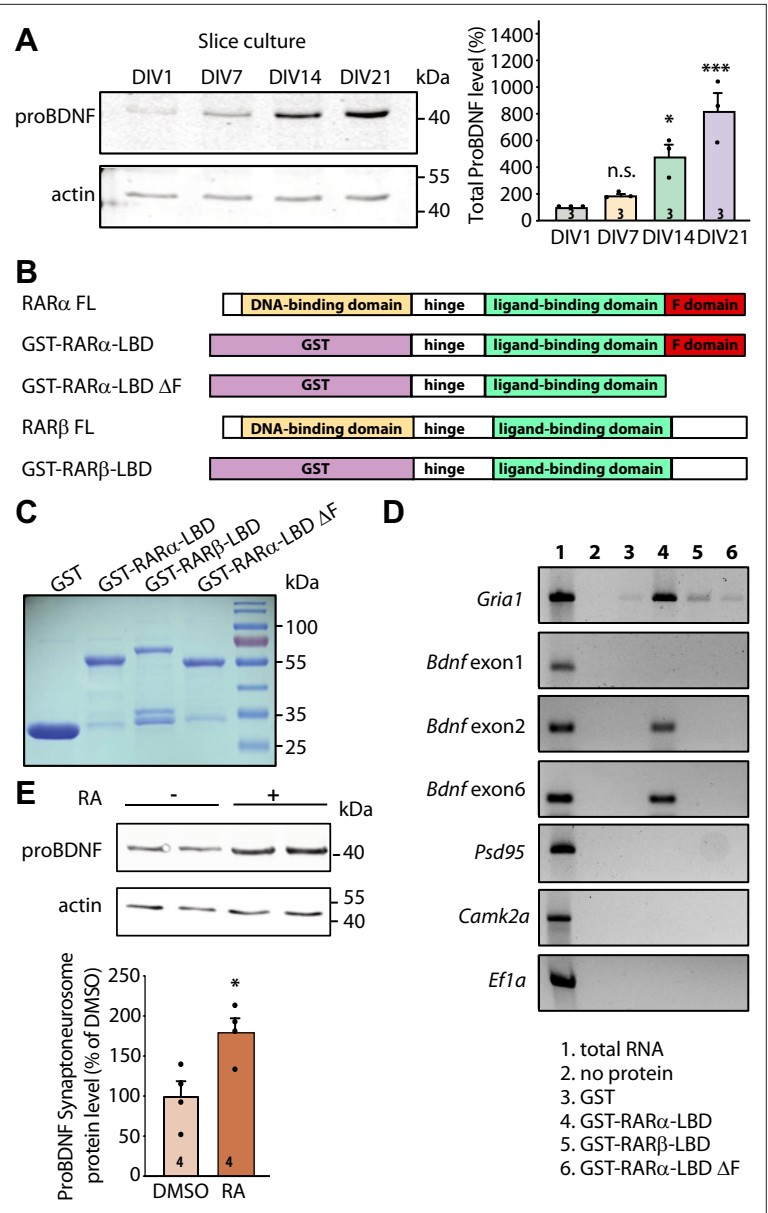

**Figure 2.** RARα binds specific brain-derived neurotrophic factor (*Bdnf*) transcript isoforms. (**A**) Representative immunoblots (left) and quantification (right) depicting proBDNF expression profiles in cultured hippocampal slices collected at 1, 7, 14, and 21 days in culture. Actin was used as a loading control and all expression levels were normalized to that of 1 day in vitro (DIV) (one-way ANOVA with Dunnett's multiple comparison test, ***, p<0.001; *, p<0.05). (B)Schematic diagram of recombinant GST fusion proteins of RARα LBD, RARα LBD ΔF, and RARβ LBD used in RNA-binding assays. Full-length RARα and RARβ protein structures are shown as references. (**C**) Representative imaging of Coomassie brilliant blue-stained SDS-polyacrylamide gel showing the expression of purified recombinant GST and GST fused RARα and RARβ LBD proteins (n=3). (**D**) Representative images for semi-quantitative RT-PCR of specific *Bdnf* transcripts pulled down from total hippocampal RNAs in in vitro selection with purified GST fusion proteins. *Gria1* mRNA was used as a positive control. *Psd95*, *Camk2a,* and *Ef1a* mRNAs served as negative controls. The representative image shown here is from one of the three experiments with similar results. (**E**) Representative immunoblot (left) and quantification (right) showing induced proBDNF synthesis in synaptoneurosomal fraction following 30 min of retinoic acid (RA) treatment. Actin was used as a loading control (two- tailed unpaired t-test, *, p<0.05). N represents number of independent experiments. All graphs represent mean ± SEM.

The online version of this article includes the following source data and figure supplement(s) for figure 2:

**Source data 1.** Individual data spreadsheet in *Figure 2A and E*.

*Figure 2 continued on next page*

*Figure 2 continued*

**Source data 2.** *Figure 2A* ProBDNF and FigureS2A ProBDNF: Immunoblots depicting proBDNF expression profile in cultured hippocampal slices.

**Figure supplement 1.** Additional data related to *Figure 2*: Mouse brain-derived neurotrophic factor (*Bdnf*) gene structure and expression; local translation of specific proteins induced by retinoic acid (RA).

**Figure supplement 1—source data 1.** Individual data spreadsheet in Fig.S2A, S2E and S2F.

**Figure supplement 1—source data 2.** *Figure 2A* ProBDNF and FigureS2A ProBDNF: Immunoblots depicting proBDNF expression profile in whole hippocampi collected from mouse pups.

transcript isoforms (*Figure 2—figure supplement 1B*). All regulatory non-coding exons yield a single identical mature BDNF protein (*Aid et al., 2007*). The complex *Bdnf* gene structure gives rise to a diverse array of *Bdnf* transcripts that can be differentially trafficked to subcellular domains and translationally regulated by distinct intra- and extracellular signals (*Wang et al., 2022*; *Lau et al., 2010*; *Song et al., 2017*). We first performed in silico analysis to search for RARα recognition motifs in non-coding *Bdnf* exons. This analysis revealed that at least two *Bdnf* exons (exon 2 and exon 6) carry RARα-binding motifs in their sequence (*Figure 2—figure supplement 1B*). *Bdnf* transcripts containing exon 2 or exon 6 are trafficked to distal dendrites and are subject to synaptic activity-dependent translational regulation (*Song et al., 2017*; *Baj et al., 2011*; *Colliva and Tongiorgi, 2021*; *Vaghi et al., 2014*; *Baj, 2016*). In cultured hippocampal pyramidal neurons, the localization of *Bdnf* exon 2 and exon 6 in distal dendrites increases dramatically as the culture matures from 7 to 18 DIV (*Baj et al., 2011*), making them potential targets for RA/RARα-mediated translational regulation around the onset time of homeostatic presynaptic changes.

RARα-mediated translational regulation in dendrites requires direct association of dendritically sorted mRNAs to RARα via specific recognition motifs (*Poon and Chen, 2008*). Specifically, the carboxyl-terminal F domain of RARα mediates direct binding to substrate mRNAs (*Figure 2B*). This mRNA-binding ability is specific to RARα as another RA receptor family member RARβ, which has a different amino acid sequence in the F domain, shows only minimal association with dendritic mRNAs (*Poon and Chen, 2008*). We first performed in vitro RNA-binding assay in which RARα LBD domain (including the F domain) was fused to GST and immobilized on glutathione Sepharose beads. GST alone, GST-tagged RARα LBD lacking F-domain (RARα LBD ΔF), and RARβ LBD were used as negative controls. Total RNAs pooled from 3- to 4-week-old whole hippocampi were used in the binding assay. GST and all fusion proteins showed comparable expression and binding to the beads (*Figure 2C* and *Figure 2—figure supplement 1C*). RT-PCR was used to test the relative enrichment of selected mRNAs by RARα. *Bdnf* transcripts containing exon 2 or exon 6 exhibited selective enrichment through binding to RARα LBD, along with *Gria1* mRNAs, an mRNA substrate of RARα identified in our previous study (*Poon and Chen, 2008*; *Figure 2D*). Transcripts of *Psd95*, *Camk2a*, and *Ef1a*, which are also dendritically localized but do not exhibit RARα binding in previous studies (*Poon and Chen, 2008*), served as negative controls (*Figure 2D*). By contrast, *Bdnf* transcript containing exon 1 that carries only a single RARα recognition motif (*Figure 2—figure supplement 1B*) and is not sorted to dendrites in hippocampal neurons (*Baj et al., 2011*) failed to bind to RARα LBD (*Figure 2D*). Additionally, deletion of F domain, the mRNA-binding domain at the C-terminal end of RARα, abolished the binding activity of all mRNAs (*Figure 2D*). Taken together, this data indicates that the two dendritically localized *Bdnf* mRNAs (carrying exon 2 and exon 6) exhibit specific binding to RARα F domain.

Does the binding of *Bdnf* mRNAs by RARα result in their translational regulation by RA? To address this, we examined RA-induced local translation of BDNF in synaptoneurosomes prepared from 3- to 4-week-old mouse hippocampi. The purity of synaptoneurosomal fraction was verified by the enrichment of synaptic protein PSD95 and the absence of nuclear protein histone H3 (*Figure 2—figure supplement 1D*). A brief treatment of synaptoneurosomes with 1 µM RA for 30 min at 37°C significantly increased GluA1 protein levels and demonstrated the translational competency of the synaptoneurosomal preparation (*Aoto et al., 2008*; *Figure 2—figure supplement 1E*). Importantly, the expression levels of proBDNF protein also increased significantly (*Figure 2E*) while PSD95, whose mRNAs are also dendritically localized but do not bind to RARα, failed to respond to RA stimulation (*Poon and Chen, 2008*; *Figure 2—figure supplement 1F*). Thus, RARα binding to *Bdnf* mRNAs does convey translational regulation of BDNF by RA.

## RA drives retrograde BDNF-TrkB signaling to achieve regulation of presynaptic homeostatic plasticity

Having established that RA regulates BDNF synthesis through direct binding between RARα and dendritically localized isoforms of *Bdnf* mRNAs, we next sought to test the relevance between RA-dependent BDNF synthesis and presynaptic homeostatic changes. To this end, we prepared organotypic hippocampal slice cultures from conditional knockout mice for either BDNF or TrkB, which allowed for dissection of the loci of action (pre vs. postsynaptic) for these two key players. Injection of Cre-expressing AAVs into either the CA3 or CA1 regions of the hippocampus achieved pre- or postsynaptic-specific deletion of target proteins in the Schaeffer collateral-CA1 synapses. Acute RA treatment or prolonged CNQX treatment significantly increased both mEPSC amplitudes and frequencies in uninfected control slices (*Figure 3A–C*). While presynaptic deletion of BDNF did not impair either of these changes in mEPSCs, postsynaptic deletion of BDNF prevented the increase in mEPSC frequency (*Figure 3B–C*). By contrast, presynaptic, but not postsynaptic deletion of TrkB blocked the increase in frequency induced by either RA or CNQX (*Figure 3D–F*). Deletion of either BDNF or TrkB pre- or postsynaptically did not affect the homeostatic increase in mEPSC amplitude, indicating that BDNF/TrkB signaling is exclusively involved in regulation of presynaptic function. The locations of action for BDNF and TrkB are consistent with the postsynaptic initiation of RA synthesis followed by RARα-mediated translational regulation supporting the notion that retrograde BDNF signaling through presynaptic TrkB drives presynaptic changes during homeostatic synaptic plasticity (*Figure 3—figure supplement 1*).

## Discussion

This study describes an RA/RARα-dependent trans-synaptic retrograde signaling pathway that modulates presynaptic function during homeostatic plasticity. Here, we identify dendritically localized postsynaptic *Bdnf* transcripts as RARα-binding targets that are subject to RA-dependent regulation of BDNF synthesis upon activity blockade. BDNF enhances presynaptic function via a retrograde signaling mechanism by binding to presynaptic TrkB receptors. In concert with an RA-dependent increase in postsynaptic AMPAR abundance, this molecular pathway contributes to homeostatic adjustment of synaptic strength during chronic activity blockade.

BDNF has diverse roles in synapse function and plasticity and is implicated in the pathophysiology of various brain disorders (*Lima Giacobbo et al., 2019*; *Bathina and Das, 2015*; *Jin et al., 2019*; *Paredes et al., 2022*; *Autry and Monteggia, 2012*). As one of the key trans-synaptic signaling molecules, the transcription, synthesis, and secretion of BDNF are highly activity-dependent (*Wong et al., 2015*; *Vermehren-Schmaedick et al., 2015*; *Miyasaka and Yamamoto, 2021*; *Kohara et al., 2001*; *Horvath et al., 2021*). This activity-dependent property of BDNF expression and secretion plays important roles in synapse development and remodeling (*Hu et al., 2005*; *Choo et al., 2017*; *Yoshii and Constantine-Paton, 2010*; *Je et al., 2013*; *Gottmann et al., 2009*; *Bamji et al., 2006*; *Wong-Riley, 2021*; *Genoud et al., 2004*; *Shen and Cowan, 2010*). Moreover, through its actions on both inhibitory and excitatory synapses, BDNF signaling impacts synaptic function (*Rauti et al., 2020*; *West, 2008*; *Lu, 2003*; *Karpova, 2014*; *Crozier et al., 2008*), activity-dependent synaptic plasticity mechanisms such as long-term potentiation and long-term depression (*Lu et al., 2008*; *Panja and Bramham, 2014*; *Aicardi et al., 2004*; *Garad et al., 2021*; *Aarse et al., 2016*; *Gangarossa et al., 2020*; *Lu et al., 2014*), and ultimately learning and memory formation (*Leal et al., 2014*; *Gonzalez et al., 2019*; *Cunha et al., 2010*; *Bekinschtein et al., 2014*; *Bekinschtein et al., 2008*). However, how synaptic inactivity regulates BDNF expression in the context of homeostatic synaptic plasticity is not yet fully understood.

The impact of exogenous BDNF on basal synaptic function and its connection to homeostatic plasticity has been investigated in several studies (*Wang et al., 2022*; *Fernandes and Carvalho, 2016*). In visual cortical pyramidal neurons, exogenous BDNF blocks homeostatic upscaling induced by chronic synaptic inactivity while BDNF depletion with TrkB-IgG scales up mEPSC amplitudes (*Rutherford et al., 1998*). While these results suggest a potential involvement of BDNF signaling in cortical pyramidal neuron homeostatic plasticity, chronic treatment of BDNF does not downscale mEPSCs (*Leslie et al., 2001*). Moreover, BDNF's effect on basal synaptic transmission seems to be cell type- and region-specific. Different from its effect on cortical pyramidal cells, BDNF increases mEPSC

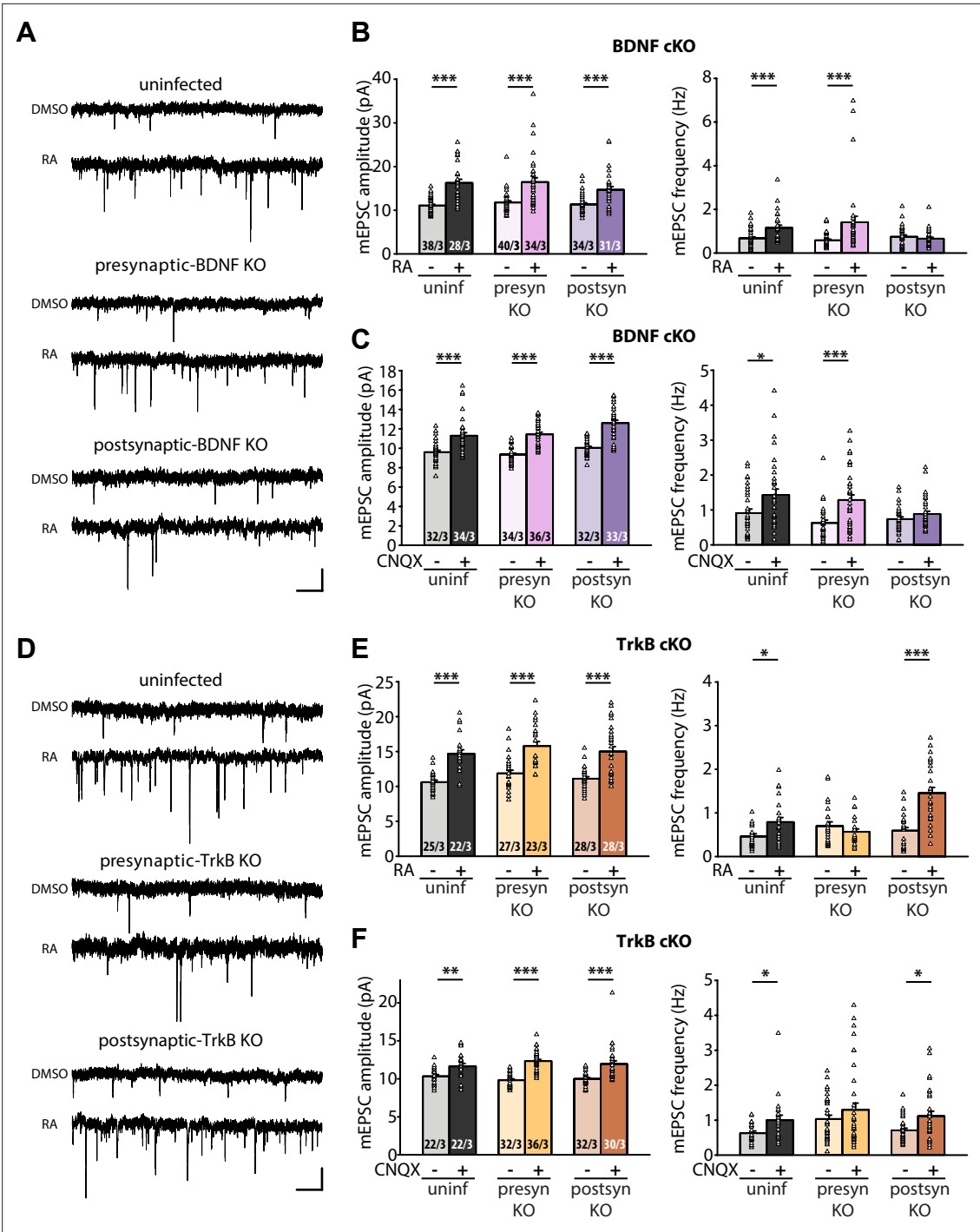

**Figure 3.** Retrograde BDNF signalling is required for RARα-mediated regulation of presynaptic homeostatic scaling. (**A**) Example traces of mEPSCs recorded from hippocampal pyramidal neurons in organotypic slices from WT (uninfected), presynaptic BDNF KO (Cre expression in CA3) and postsynaptic BDNF KO (Cre expression in CA1) groups treated with DMSO or RA (10 μM, 4 hr). Scale bars, 10 pA, 0.5 sec. (**B**) Quantification of mEPSC amplitudes and frequencies recorded from WT, presynaptic and postsynaptic BDNF KO neurons treated with DMSO or RA. ***, $p < 0.001$; two-way ANOVA followed by Mann Whitney test. Amp: $F_{(2,199)} = 1.062$, $p > 0.3$; freq: $F_{(2,199)} = 6.244$, $p < 0.01$. (**C**) Quantification of mEPSC amplitudes and frequencies recorded from WT, presynaptic and postsynaptic BDNF KO neurons treated with DMSO or CNQX (36 hours). *, $p<0.05$, **, $p < 0.01$; ***, $p < 0.001$; two-way ANOVA followed by Mann Whitney test. Amp: $F_{(2,195)} = 1.77$, $p > 0.15$; freq: $F_{(2,195)} = 2.53$, $p >0.05$. (**D**) Example traces of mEPSCs recorded from hippocampal pyramidal neurons in organotypic slices from WT (uninfected), presynaptic TrkB KO (Cre expression in CA3) and postsynaptic TrkB KO (Cre expression in CA1) groups treated with DMSO or RA (10 μM, 4 hr). Scale bars, 10 pA, 0.5 sec. (**E**) Quantification of mEPSC amplitudes and frequencies recorded from WT, presynaptic and postsynaptic TrkB KO neurons treated with DMSO or RA. **, $p < 0.01$; ***, $p$

*Figure 3 continued on next page*

**Figure 3 continued**

< 0.001; two-way ANOVA followed by Mann Whitney test. Amp: $F_{(2,147)} = 0.01$, $p > 0.9$; freq: $F_{(2,147)} = 14.87$, $p < 0.0001$. (**F**) Quantification of mEPSC amplitudes and frequencies recorded from WT, presynaptic and postsynaptic TrkB KO neurons treated with DMSO or CNQX (36 hours). *, p<0.05,**, $p < 0.01$; ***, $p < 0.001$; two-way ANOVA followed by Mann Whitney test. Amp: $F_{(2,168)} = 2.33$, $p > 0.1$; freq: $F_{(2,168)} = 0.17$, p> 0.8.n/N represent number of neurons/number of independent experiments. All graphs represent mean ± SEM.

The online version of this article includes the following source data and figure supplement(s) for figure 3:

**Source data 1.** Individual data spreadsheet in *Figure 3B, C, E and F*.

**Figure supplement 1.** A working model for retinoic acid (RA)-mediated regulation of presynaptic and postsynaptic homeostatic plasticity.

amplitudes in visual cortical interneurons (***Rutherford et al., 1998***). In the hippocampus, BDNF enhances mEPSC amplitudes without affecting their frequencies, but increases mIPSC frequency and sizes of GABAergic synaptic terminals (***Bolton et al., 2000***). In nucleus accumbens medium spiny neurons, acute BDNF treatment (30 min) increases surface AMPAR expression, but chronic BDNF treatment (24 hr) decreases surface AMPAR expression and blocks synaptic downscaling (***Reimers et al., 2014***; ***Li and Wolf, 2011***). Taken together, it becomes apparent that BDNF signaling is complex as its effect on synaptic transmission is multifaceted and dependent on the context of the study (cell type, brain region, developmental stage, etc.) (***Wang et al., 2022***).

In addition to studies examining the impact of exogenous BDNF on basal synaptic transmission, the involvement of endogenous BDNF signaling in homeostatic synaptic plasticity has been further investigated and found to be required for both synaptic upscaling and downscaling in hippocampal neurons (***Lindskog et al., 2010***; ***Jakawich et al., 2010***; ***Horvath et al., 2021***). In the case of synaptic hyperactivity-induced synaptic downscaling, mIPSC blockade and postsynaptic increase in calcium influx were found to induce postsynaptic increase in *Bdnf* transcription, which mediates homeostatic down-regulation of mEPSC amplitudes (***Horvath et al., 2021***). Paradoxically, in the case of synaptic upscaling, postsynaptic release of BDNF as a retrograde messenger was also required in synaptic activity blockade-induced homeostatic increase in presynaptic vesicle turnover (***Lindskog et al., 2010***) and mEPSC frequency (***Jakawich et al., 2010***). Furthermore, rapid activation of dendritic mTORC1 signaling was proposed to be a molecular mechanism for protein synthesis-dependent release of postsynaptic BDNF to mediate presynaptic homeostatic compensation under prolonged inactivation of postsynaptic AMPARs (***Henry et al., 2012***; ***Henry et al., 2018***). Interestingly, at the *Drosophila* neuromuscular junction, where robust homeostatic increase in presynaptic release occurs in response to reduced postsynaptic responsiveness, postsynaptic TOR activation is also required for the retrograde signaling that initiates the presynaptic changes (***Penney et al., 2012***; ***Goel et al., 2017***). Thus, although retrograde messenger systems participating in presynaptic homeostatic plasticity are likely distinct between vertebrate and invertebrate synapses, they nonetheless share a common requirement of upstream TOR/mTOR-dependent translational activation in the postsynaptic compartment, and a common functional outcome – the enhanced presynaptic release probability. Does the mTOR-dependent translational regulation acts in parallel to the RARα-dependent pathway? A previous study unexpectedly discovered that RARα clamps mTOR's activity level during neuronal activation through inhibition of ERK (***Hsu et al., 2019***). Thus, it remains to be explored as how the RA/RARα- and the mTOR-dependent pathways interact/integrate in the context of synaptic plasticity to achieve a concerted adjustment of pre- and postsynaptic changes.

How does the same synaptic signaling molecule (BDNF) act in seemingly opposite biological processes? The vastly diverse functions of BDNF in the nervous system have been attributed to tight spatial, temporal, and stimulus-dependent regulation of BDNF expression from its complex gene structure. Using a combination of individual promoters (served by initial eight non-coding exons in rodents) and polyadenylation sites, a single *Bdnf* gene could give rise to several different transcripts (***Aid et al., 2007***). Although it is widely accepted that most of these individual *Bdnf* promoters respond differentially to neuronal activity, the specific functional roles of most of these transcripts remain elusive. For instance, multiple studies have shown that promoter/exon IV is robustly involved in activity-dependent BDNF expression in the cortex (***Timmusk et al., 1993***; ***Zheng et al., 2011***; ***Sakata et al., 2013***; ***Martinowich et al., 2011***; ***Ratnu et al., 2014***; ***Hong et al., 2008***). In addition, BDNF expression driven by promoter/exon I, III, and VI has been shown to be regulated by AP1 transcription factors and by TrkB signaling in a positive feedback loop (***West, 2008***; ***Tuvikene et al., 2016***). However, how synaptic inactivity modulates specific exon-derived BDNF expression

and which regulatory exons/promoters of the *Bdnf* gene respond to synaptic inactivity has never been explored. Further, how specific *Bdnf* transcripts contribute to homeostatic scaling is largely unknown. Our study focuses on the involvement of different *Bdnf* transcripts in synaptic inactivity-induced presynaptic upscaling. We found that dendritically localized *Bdnf* transcripts containing exon II and VI are subjected to RA/RARα-mediated regulation of synthesis in the postsynaptic compartment under chronic synaptic activity blockade. These newly synthesized BDNF acts retrogradely on presynaptic TrkB receptors to mediate homeostatic modulation of presynaptic function. An increase in *Bdnf* transcription and enhanced dendritic targeting of specific *Bdnf* transcripts have been associated with increased neuronal and synaptic activity (*Miyasaka and Yamamoto, 2021*; *Horvath et al., 2021*; *Shimada et al., 1998*; *Tongiorgi et al., 1997*; *An et al., 2008*; *Verpelli et al., 2010*; *Tanaka et al., 2008*). Our study shows that by acting on specific dendritically targeted *Bdnf* transcripts, RA/RARα selectively activates BDNF synthesis in postsynaptic compartments during synaptic inactivity, thus further extending the molecular regulatory mechanisms underlying BDNF signaling in synaptic inactivity-induced synaptic plasticity.

Similar to the rodent *Bdnf* gene, the human *BDNF* gene has a complex structure with multiple transcription initiation sites and alternative transcripts (*Pruunsild et al., 2007*), and is implicated in neuropsychiatric disorders (*Wang et al., 2022*; *Autry and Monteggia, 2012*; *Cattaneo et al., 2016*). The involvement of synaptic RA/RARα signaling in various neuropsychiatric diseases has begun to emerge in recent years (*Crofton et al., 2021*; *Zhang et al., 2019*; *Zhang et al., 2016*; *Suzuki, 2021*). In particular, the absence of RA-RARα-regulated homeostatic plasticity mechanisms have been established in both *Fmr1* knockout mice (a mouse model for fragile X syndrome [FXS]) and human neurons differentiated from FXS patient-derived induced pluripotent stem cells (*Soden and Chen, 2010*; *Zhang et al., 2013*; *Zhong et al., 2018*). The convergence of BDNF signaling and RA signaling for synaptic functions and homeostatic plasticity highlights the importance of understanding the intersections of various synaptic signaling pathways and their implications in cognitive function.

# Materials and methods

**Key resources table**

| Reagent type (species) or resource | Designation | Source or reference | Identifiers | Additional information |
|---|---|---|---|---|
| Strain, strain background (*Mus musculus*) | *Rara^fl/fl* | *Chapellier et al., 2002* | N/A | |
| Strain, strain background (*Mus musculus*) | *Bdnf^fl/fl* | *Rios et al., 2001* (The Jackson Lab) | JAX#004339 RRID: IMSR_ JAX:004339 | |
| Strain, strain background (*Mus musculus*) | *Ntrk2^fl/fl* | *Luikart, 2005* | N/A | |
| Strain, strain background (*Mus musculus*) | CD-1 IGS | Charles River Laboratories | Strain code: 022 | |
| Antibody | Anti-Glutathione-*S*-transferase (GST) (Mouse monoclonal) | Sigma | SAB4200237-200UL; Clone 2H3-D10 | WB (1:1000) |
| Antibody | Anti-BDNF (Rabbit monoclonal) | Abcam | ab108319 | WB (1:1000) |
| Antibody | Anti-Glur1-NT (N-terminus) (Mouse monoclonal) | Millipore | MAB2263; Clone RH95 | WB (1:2000) |
| Antibody | Anti-PSD95 (Mouse monoclonal) | Invitrogen | MA1-046 | WB (1:1000) |
| Antibody | Anti-Histone H3 (Rabbit polyclonal) | Millipore | 07-690 | WB (1:5000) |

*Continued on next page*

*Continued*

| Reagent type (species) or resource | Designation | Source or reference | Identifiers | Additional information |
|---|---|---|---|---|
| Antibody | Anti-Actin (Mouse monoclonal) | Millipore | MAB1501; Clone C4 | WB (1:5000) |
| Antibody | IRDye 800CW IgG Secondary Antibody (Donkey anti-Rabbit) | Li-cor | P/N: 926-32213 | WB (1:10,000) |
| Antibody | IRDye 680RD IgG Secondary Antibody (Donkey anti-Mouse) | Li-cor | P/N: 926–68072 | WB (1:10,000) |
| Recombinant DNA reagent | pGEX-KG | | | Expression vector with GST tag |
| Recombinant DNA reagent | pGEX-KG-RARα LBD | This paper | | Nucleotides 460 to end |
| Recombinant DNA reagent | pGEX-KG-RARα LBD ΔF | This paper | | Nucleotides 460–1,251 |
| Recombinant DNA reagent | pGEX-KG- RARβ LBD | This paper | | 496 to end |
| Sequence-based reagent | *Gria1*_F | This paper | PCR primers | caatcacaggaacatgcggc |
| Sequence-based reagent | *Gria1*_R | This paper | PCR primers | cctgccagttcttctcggcggc |
| Sequence-based reagent | Exon 1 *Bdnf* _F | This paper | PCR primers | ctccctcactttctctggg |
| Sequence-based reagent | Exon 1 *Bdnf* _R | This paper | PCR primers | ctgagagacacgtttccc |
| Sequence-based reagent | Exon 2 *Bdnf* _F | This paper | PCR primers | cgagccccagtttggtcccc |
| Sequence-based reagent | Exon 2 *Bdnf* _R | This paper | PCR primers | ggtggctagatcctggtg |
| Sequence-based reagent | Exon 6 *Bdnf* _F | This paper | PCR primers | gacccggttccttcaactgcc |
| Sequence-based reagent | Exon 6 *Bdnf* _R | This paper | PCR primers | ctcagggtccacacaaagctctcgg |
| Sequence-based reagent | *Dlg4 (Psd95)*_F | This paper | PCR primers | catcgaaggaggcgctgccc |
| Sequence-based reagent | *Dlg4 (Psd95)*_R | This paper | PCR primers | cattgtccaggtgctgagaata |
| Sequence-based reagent | *Camk2a*_F | This paper | PCR primers | cattgtggcccgggagtatt |
| Sequence-based reagent | *Camk2a*_R | This paper | PCR primers | ggtgatgggaaatcataggcacc |
| Sequence-based reagent | *Ef1a*_F | This paper | PCR primers | cgagaccagcaaatactatgtgacc |
| Peptide, recombinant protein | *Ef1a*_R | This paper | PCR primers | ggcatattagcacttggctcc |
| Commercial assay or kit | PrimeScript RT Reagent Kit with gDNA Eraser (Perfect Real Time) | Takara | RR047B | |
| Chemical compound, drug | Retinoic acid (RA) | Sigma | R2625-50MG | |

*Continued on next page*

*Continued*

| Reagent type (species) or resource | Designation | Source or reference | Identifiers | Additional information |
|---|---|---|---|---|
| Chemical compound, drug | CNQX | Tocris | 0190 | |
| Chemical compound, drug | cOmplete, EDTA-free Protease Inhibitor Cocktail | Sigma | 4693132001 | |
| Chemical compound, drug | RNasin Ribonuclease Inhibitors | Promega | N2115 | |
| Chemical compound, drug | TRIzol Reagent | Thermo Fisher Scientific | 15596026 | |
| Software, algorithm | Image Studio 5.2.5 | LI-COR Bioscience | https://www.licor.com/bio/products/software/image_studi o_lite/ | |
| Software, algorithm | Prism 9 | GraphPad | https://www.graphpad.com/scientific-software/prism/ | |
| Software, algorithm | ImageJ | NIH | https://imagej.nih.gov/ij/ | |
| Software, algorithm | Mind the graph | | https://mindthegraph.com/ | |
| Other | Glutathione Sepharose 4B | Sigma Millipore | GE17-0756-05 | In vitro RNA-binding assay |

## Animals

Mouse strains used in the study are mentioned in table above. All mouse studies were performed according to protocols approved by the Stanford University Administrative Panel on Laboratory Animal Care. All procedures conformed to NIH Guidelines for the Care and Use of Laboratory Animals and were approved by the Stanford University Administrative Panel.

## Plasmid constructs for recombinant gene expression

All RAR constructs were generated using mouse sequences and cloned into pGEX-KG. For GST-RARα LBD, nucleotides 460 to end were cloned using BamHI and HindIII; for GST-RARα LBD ΔF, nucleotides 460–1251 were cloned with BamHI and HindIII; and for GST-RARβ LBD, nucleotides 496 to end were cloned using BamHI and HindIII. All plasmids generated in this study are available upon request.

## In vitro RNA-binding assay

Purified GST fusion proteins and total RNA from whole hippocampi of P25-P30 mouse pups were used for selection. GST fusion proteins were expressed in BL21 cells induced with 1 mM IPTG. Bacteria was sonicated in lysis buffer (150 mM NaCl, 20 mM sodium phosphate, pH 7.4, 1% Triton X-100, and protease inhibitors) and debris were cleared by centrifugation at 10,000 rpm for 20 min. Protein expression was confirmed by SDS/PAGE followed by Coomassie staining and immunoblotting when possible. GST fused to the various domains of RARα or RARβ were then purified from the bacterial lysate by binding to glutathione Sepharose beads and equilibrating/washing the protein-bound beads five times in RNA-binding buffer (200 mM KOAc, 10 mM TrisOAc, pH 7.7, and 5 mM MgOAc with protease and RNase inhibitors). Total RNA was obtained from whole hippocampi of 25- to 30-day-old CD1 mice using TRIzol. RNA was DNase treated and reextracted with TRIzol, and the pellet was resuspended in nuclease-free water and quantified by spectrophotometry. RNA (20 μg) was added to RNA-binding buffer and heated to 95°C to denature secondary structure, then slowly renatured. Renatured RNA was then added to the immobilized GST fusion protein in RNA-binding buffer and rotated overnight at 4°C. Beads were then washed several times in RNA-binding buffer. RNA was extracted with TRIzol, treated with RNase free DNase I, then reverse transcribed with oligo(dT) according to the manufacturer's instructions. cDNA was used for amplification with PCR using gene-specific primers.

## Synaptoneurosome preparation

Hippocampi from P25-P30 CD1 mice were dissected and gently homogenized in a solution containing 33% sucrose, 10 mM HEPES, 0.5 mM EGTA (pH 7.4), and protease inhibitors. Nuclei and other debris were pelleted at 2,000 × *g* for 5 min at 4°C and the supernatant was filtered through three

layers of 100 µm pore nylon mesh (Millipore), and a 5 µm pore PVDF syringe filter (Millipore). The filtrate was then centrifuged for 10 min at 10,000 × *g* at 4°C and the supernatant was removed. The synaptoneurosome-containing pellet was then resuspended in the appropriate amount of lysis buffer containing 140 mM NaCl, 3 mM KCl, 10 mM glucose, 2 mM $MgSO_4$, 2 mM $CaCl_2$, and 10 mM HEPES (pH 7.4) containing protease inhibitors and RNAsin. Equal volumes were then aliquoted into opaque microfuge tubes. Appropriate samples were incubated with 1 µM RA for 10 min at 37°C and immediately frozen in dry ice afterward.

## Western blotting

Samples were run on 10% SDS/PAGE and transferred to nylon membranes. Membranes were blocked with Tris-buffered saline solution containing 0.1% Tween-20 (TBST) and 5% dry milk. Primary antibodies were diluted into TBST and incubated overnight at 4°C. Primary antibody was washed with TBST and secondary antibody was added for 1 hr in TBST. Secondary antibody was washed off with TBST and signal detected using Odyssey CLx imaging system (LI-COR). The densitometric analysis was performed using ImageJ software.

## Organotypic hippocampal slice cultures

Organotypic slice cultures were prepared from postnatal day 7–8 mouse pups and placed on semiporous membranes (Millipore) for 21–25 days prior to recording (*Gähwiler et al., 1997*). Briefly, slices were maintained in a MEM-based culture media comprised of 1 mM $CaCl_2$, 2 mM $MgSO_4$, 1 mM L-glutamine, 1 mg/L insulin, 0.0012% ascorbic acid, 30 mM HEPES, 13 mM D-glucose, and 5.2 mM $NaHCO_3$. Culture media was a pH of 7.25 and the osmolarity was 320. Cultures were maintained in an incubator with 95% $O_2$/5% $CO_2$ at 34°C.

## Viral vectors and viral infection

AAV Syn-CRE was prepared as previously described (*Aoto et al., 2013*). Viral titers were determined by qPCR.

Cultures were injected on 0 DIV and maintained for 21–25 days prior to recording. Presynaptic deletion was accomplished via injection of AAV-CRE into the CA3 pyramidal cell body layer. Postsynaptic deletion was achieved via injection of AAV-CRE into the CA1 pyramidal cell body layer. All experiments are executed with interweaving controls (either uninfected [i.e. WT], presynaptic deletion, or postsynaptic deletion) All injections were verified and confirmed as 95–100% infectivity in either the CA3 or CA1 prior to recording.

## Electrophysiology

Voltage-clamp whole-cell recordings are obtained from CA1 pyramidal neurons treated with either vehicle controls,10 µM RA for 4 hr prior to recording, or 20 µM CNQX for 36 hr prior to recording, under visual guidance using transmitted light illumination. Vehicle control, RA treated, and CNQX-treated cells were obtained from the same batches of slices on the same experimental day.

The recording chamber is perfused with 119 mM NaCl, 2.5 mM KCl, 4 mM $CaCl_2$, 4 mM $MgCl_2$, 26 mM $NaHCO_3$, 1 mM $NaH_2PO_4$, 11 mM glucose, and 0.1 mM picrotoxin, at pH 7.4, gassed with 5% $CO_2$/95% $O_2$ and held at 30°C. Patch recording pipettes (3–6 MOhm) are filled with 115 mM cesium methanesulfonate, 20 mM CsCl, 10 mM HEPES, 2.5 mM $MgCl_2$, 4 mM $Na_2ATP$, 0.4 mM $Na_3GTP$, 10 mM sodium phosphocreatine, and 0.6 mM EGTA at pH 7.25.

Spontaneous miniature transmission was obtained in the presence of 1 µM of TTX in the external solution. For slices previously exposed to RA or CNQX, slices were washed out prior to recording spontaneous responses.

## Statistical analyses

Sample sizes were determined based on power analysis (power set at 0.8 and α=0.05) and also on similar experiments performed and published by our lab and others previously (*Poon and Chen, 2008*; *Soden and Chen, 2010*; *Arendt et al., 2015*; *Henry et al., 2018*). All graphs represent average values ± SEM. Statistical differences were calculated according to parametric or nonparametric tests (indicated in figure legends). Comparisons between multiple groups were performed either with one-way ANOVA or with the two-way ANOVA followed by Tukey's test. When significant differences were

observed, p values for pairwise comparisons were calculated according to two-tailed Mann-Whitney tests (for unpaired data) or Wilcoxon tests (for paired data). Comparisons between cumulative distributions were performed according to two-sample Kolmogorov-Smirnov tests. p values are indicated in each figure.

## Acknowledgements

We thank Drs Xiling Li, Michelle Tjia, and Omid Miry for their helpful input and technical support. The work was supported by NIH grants MH086403 (LC), NS11566001 (LC), HD104458 (LC).

## Additional information

### Competing interests

Lu Chen: Senior editor, eLife. The other authors declare that no competing interests exist.

### Funding

| Funder | Grant reference number | Author |
| --- | --- | --- |
| National Institute of Mental Health | MH086403 | Lu Chen |
| National Institute of Neurological Disorders and Stroke | NS11566001 | Lu Chen |
| Eunice Kennedy Shriver National Institute of Child Health and Human Development | HD104458 | Lu Chen |

The funders had no role in study design, data collection and interpretation, or the decision to submit the work for publication.

### Author contributions

Shruti Thapliyal, Data curation, Formal analysis, Validation, Investigation, Visualization, Methodology, Writing - original draft; Kristin L Arendt, Conceptualization, Data curation, Formal analysis, Methodology, Writing – review and editing; Anthony G Lau, Conceptualization, Data curation, Formal analysis; Lu Chen, Conceptualization, Supervision, Funding acquisition, Visualization, Methodology, Project administration, Writing – review and editing

### Author ORCIDs

Shruti Thapliyal http://orcid.org/0000-0003-3531-3018
Lu Chen https://orcid.org/0000-0002-8097-2699

### Ethics

All mouse studies were performed according to protocols approved by the Stanford University Administrative Panel on Laboratory Animal Care (#29679) . All procedures conformed to NIH Guidelines for the Care and Use of Laboratory Animals and were approved by the Stanford University Administrative Panel.

### Decision letter and Author response

Decision letter https://doi.org/10.7554/eLife.79863.sa1
Author response https://doi.org/10.7554/eLife.79863.sa2

## Additional files

### Supplementary files
• MDAR checklist

**Data availability**

All data generated and analyzed in this study are included in the manuscript and supporting files; the source data files contain the numerical data and original images used to generate the figures.

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
