## [Editor Report]

This study defines an important homeostatic pathway at excitatory synapses where retinoic acid-dependent signaling in response to synaptic inactivity drives the local synthesis of the neurotrophin BDNF in dendrites, which in turn is released to adaptively modify neurotransmitter release properties at presynaptic terminals. The authors use genetic tools to localize the action of distinct components of the pathway to pre- vs post-synaptic compartments and use biochemical approaches to define a molecular link between RA and the local translation of specific BDNF transcripts. The experiments have been well-executed and the compelling findings fill a gap in our knowledge about how presynaptic function is adaptively modulated by retrograde BDNF signaling by highlighting the role of RA in this process.

---

## [Decision Letter]

**Decision letter after peer review:**

Thank you for submitting your article "Retinoic acid-gated BDNF synthesis in neuronal dendrites drives presynaptic homeostatic plasticity" for consideration by *eLife*. Your article has been reviewed by 3 peer reviewers, including Dion K Dickman as Reviewing Editor and Reviewer #1, and the evaluation has been overseen by Gary Westbrook as the Senior Editor. The following individual involved in review of your submission has agreed to reveal their identity: Michael A Sutton (Reviewer #2).

The reviewers have discussed their reviews with one another, and the Reviewing Editor has drafted the Essential Revisions section to help you prepare a revised submission.

Essential revisions:

This manuscript probes the mechanism of postsynaptic retinoic acid (RA) signaling on presynaptic function. BDNF has important roles in synaptic plasticity, but how retrograde BDNF signaling is controlled following synaptic inactivity is unclear. The authors use genetic tools to localize the action of different components of the pathway to pre- or post-synaptic compartments and use biochemical approaches to define a molecular link between retinoic acid and local translation of distinct BDNF transcripts. The findings presented here fill a gap in our knowledge regarding how presynaptic function is adaptively modulated by BDNF by highlighting the role of RA in this process. The experiments have been well-executed and the data provide compelling support for the model proposed by the authors. There are, however, some important issues to address:

1) Evaluate the relative infection efficiency in CA3 neurons: This is critical in interpreting the negative result with RARa deltion. The pre-synaptic Cre expression would require a high percentage of CA3 neurons infected. From Figure 3, it appears this is achieved, but some information on this point would be helpful. Also in the results text for Figure 1, it should mention the use of AAV to express Cre (this is mentioned for Figure 3 but not Figure 1).

2) Define and discuss what the change in mEPSC frequency means: Changes in mEPSC frequency may reflect multiple synaptic adaptations – changes in presynaptic release probability or alterations in synapse numbers. The experiments done clearly delineate pre- and/or postsynaptic roles for RARα, BDNF, and TrkB, but alone, mEPSC frequency changes could reflect either enhanced release probability, growth of new synaptic contacts, or AMPAR recruitment to silent synapses. The authors should explicitly address these potential alternative interpretations early on in the paper. There is substantial evidence from other studies that the mEPSC frequency changes under these conditions reflect altered presynaptic release as the authors suggest, and making contact with this literature would make the conclusions of this paper that much more compelling.

3) Discuss discrepancies with previous studies: The issue of age (of culture and animals) on whether this form of pre-synaptic plasticity occurs is important (although was also established in the earlier Wang paper). Some papers have reported frequency changes during HSP but many have reported no changes (including several from the Chen lab). Age/DIV seems to explain at least some of the difference, but I haven't gone through all the papers to check. It does seem that BDNF levels are increased even by 14DIV for slice cultures (Figure 2A). What age of slice cultures were used in early papers (like Aoto, 2008; this information was missing from that paper)? This has been a contentious issue over the years, and since this offers a potential resolution to the discrepancies, more discussion of it is warranted.

4) Assess mIPSC changes: It is noted in the discussion that BDNF can increase mIPSC frequency. Does this happen here? During HSP, at least post-synaptically, excitatory and inhibitory synapses are inversely regulated. But it is possible miniature frequency is increased for both types. Or maybe not – either way, it would be interesting to define. Perhaps mIPSC amplitude goes down, but maybe that also hasn't been shown at these older ages/DIV. Looking at mIPSCs would extend impact and improve the paper.

5) Improved discussion and integration with previous work in the field: The results demonstrate a role for RA in controlling local dendritic BDNF synthesis and suggest RARα binding as a mechanism. As the authors note, previous work has similarly implicated the mTORC1 signaling pathway in local BDNF translation in response to synaptic inactivity. Parallel findings have been observed at the *Drosophila* neuromuscular junction. The authors may want to speculate in the discussion on whether these two pathways work together or in parallel during homeostatic plasticity.

6) Respond to several points of clarification:

– Figures 1 and 3 include mEPSC traces where the background has been sharded to color coordinate with the respective histograms. This was likely done to make the figures easier to read, but the shading is somewhat distracting and really does the opposite. This is not a major issue as it does not affect the data or its interpretation, but the figures would be improved if either the traces themselves are colored or the shading is simply removed.

– The authors should consider superimposing the individual data points over the histograms for the electrophysiology data shown in Figure 1B-C and Figure 3B-F

– The model shown in Figure 3 —figure supplement 1 very nicely summarizes the RA pathway, but could also be used to include other work in the field that implicates other pathways (e.g., the mTORC1 pathway).

– In the introduction, the phrase 'runaway tendency' is vague and not obvious as to its meaning. Perhaps another phrase would be better.

– The figure labels could be improved. Pre-KO and post-KO can be misinterpreted –

"Pre-KO" could be misinterpreted as being before the Cre expression manifested. Maybe "pre-syn KO" and "post-syn KO" would be easier to understand when initially looking at the figures?

– The e-phys figures probably should show the data points (in the modern style) instead of just being bar graphs. And shouldn't the ephys statistics be 2-way ANOVAs instead of just t-tests? This likely won't impact the results but would be better.

*Reviewer #1 (Recommendations for the authors):*

Overall these findings fill a gap in our knowledge regarding how presynaptic function is adaptively modulated following postsynaptic inhibition and the connection between RA and BDNF in this process. The experiments in Figure 2 are well controlled and provide compelling evidence that RARalpha binds to BDNF mRNA transcripts, while Figures 1 and 3 localize activity in pre- vs post-synaptic compartments. A plausible model is presented in Figure S3 to describe how chronic inactivity at synapses initiates retrograde homeostatic signaling through RA and BDNF. This is a relatively brief manuscript and a question is raised about whether the specific results regarding RARalpha binding to BDNF mRNA to provide a sufficient enough advance beyond what was already known. However, the real advance here is in connecting RA to the previous work on retrograde BDNF signaling at presynaptic terminals, and separating this from what is known about postsynaptic control of AMPAR synthesis and receptor scaling. This provides an important foundation to understand the signaling systems that orchestrate pre- and post-synaptic responses to synaptic inactivity. Below are some important questions and experiments for the authors to consider.

1. Overall, the authors provide very good controls, both positive and negative, for BDNF mRNA binding to RARalpha. That being said, it seems a good control could be performed using their synaptoneurosome preparation: Add RA to synaptoneurosome preparations from the pre vs. postsynaptic RARalpha cKOs in Figure 1 to determine whether BDNF is still increased. This would be a great confirmation that postsynaptic RARalpha is necessary.

2. Can the authors further refine and/or separate the postsynaptic actions of glutamate receptor blockade/chronic inactivity between RA/RARalpha-mediated responses protein synthesis more generally? In terms of the presynaptic plasticity studied in this manuscript, both RA and BDNF seem to function in the same pathway, requiring modulation of BDNF protein synthesis by RARalpha. This retrograde signaling by BDNF is clearly distinct from the pathways that control postsynaptic homeostatic responses to inactivity, such as AMPAR synthesis and receptor scaling. Can the authors determine or define what responses are protein synthesis-dependent, RA/RARalpha-independent in the postsynaptic compartment?

3. Do the authors have any insights into what the increase in mEPSC frequency actually means for how presynaptic function is changing due to retrograde BDNF signaling gated by RA?

*Reviewer #2 (Recommendations for the authors):*

This paper examines the role of retinoic acid (RA) in regulating feedback control of presynaptic neurotransmitter release via postsynaptic synthesis and release of brain-derived neurotrophic factor (BDNF). The authors show that deletion of RA receptor α (RARα) in CA1 neurons of organotypic slices prevents the increase in mEPSC frequency that accompanies RA administration or chronic block of AMPA receptors in these cells. Importantly, deletion in presynaptic CA3 neurons has no effect, indicating a postsynaptic role for RARα. The authors then demonstrate that RARα (but not RARβ) binds specific BDNF transcripts (containing exons 2 and 6) that have been shown by previous studies to localize to distal dendrites, and that RA administration drives BDNF synthesis in isolated synaptic fractions (synaptoneurosomes). Finally, through conditional deletion experiments targeting either the pre- or postsynaptic compartment, the authors demonstrate a postsynaptic role for BDNF and a presynaptic role for its receptor TrkB in presynaptic compensation driven by RA or AMPAR blockade, revealing a transsynaptic feedback mechanism requiring postsynaptic release of BDNF and presynaptic BDNF-TrkB signaling.

The experiments have all been well executed with appropriate controls and the data provide compelling support for the conclusions of the paper. These results are important, as they reveal a homeostatic control mechanism regulating presynaptic function mediated by RA signaling that appears to operate in parallel with the well-established postsynaptic compensation mechanism first described by this group and validated by others. The findings have broad implications for our understanding of homeostatic regulation at synapses and are of significant general interest. I have just a few suggestions for the authors to consider in revising their paper.

1) Strictly speaking, changes in mEPSC frequency may reflect multiple synaptic adaptations – changes in presynaptic release probability or alterations in synapse numbers. The experiments done clearly delineate pre- and/or postsynaptic roles for RARα, BDNF, and TrkB, but alone, mEPSC frequency changes could reflect either enhanced release probability, growth of new synaptic contacts, or AMPAR recruitment to silent synapses. The authors should explicitly address these potential alternative interpretations early on in the paper. There is substantial evidence from other studies that the mEPSC frequency changes under these conditions reflect altered presynaptic release as the authors suggest, and making contact with this literature would make the conclusions of this paper that much more compelling.

2) The results demonstrate a role for RA in controlling local dendritic BDNF synthesis and suggest RARα binding as a mechanism. As the authors note, previous work has similarly implicated the mTORC1 signaling pathway in local BDNF translation in response to synaptic inactivity. The authors may want to speculate in the discussion on whether these two pathways work together or in parallel during homeostatic plasticity.

3) Figures 1 and 3 include mEPSC traces where the background has been sharded to color coordinate with the respective histograms. While I understand this was done to make the figures easier to read, the shading is somewhat distracting and really does the opposite. This is not a major issue as it does not affect the data or its interpretation, but I think the figures would be improved if either the traces themselves are colored or the shading is simply removed.

4) If possible, the authors should consider superimposing the individual data points over the histograms for the electrophysiology data shown in Figure 1B-C and Figure 3B-F.

5) The model shown in Figure 3 —figure supplement 1 very nicely summarizes the RA pathway, but could also be used to include other work in the field that implicates other pathways (e.g., the mTORC1 pathway).

*Reviewer #3 (Recommendations for the authors):*

The main issue is one of novelty. The basic findings have been made before, with this lab already showing the activity blockade increase in mEPSC frequency is RA dependent (and age dependent), and other papers (cited in the text but maybe underemphasized; Jakawich and Lindskog) have shown that this increase in frequency was BDNF dependent, released post-synaptically and signaling pre-synaptically. And the RAR is a known transcriptional regulator. This paper nicely links these elements together but it is confirming what is expected. The specific BDNF transcript involved (exon II and IV) is the novel finding.

However, it is noted that in the discussion that BDNF can increase mIPSC frequency. Does this happen here? During HSP, at least post-synaptically, excitatory and inhibitory synapses are inversely regulated. But it is possible miniature frequency is increased for both types. Or maybe not – either way, it would be interesting. I would assume that mIPSC amplitude would go down, but maybe that also hasn't been shown at these older ages/DIV. Looking at mIPSCs would add novelty and improve the paper.

The issue of age (of culture and animals) on whether this form of pre-synaptic plasticity occurs is important (although was also established in the earlier Wang paper). Some papers have reported frequency changes during HSP but many have reported no changes (including several from the Chen lab). Age/DIV seems to explain at least some of the difference, but I haven't gone through all the papers to check. It does seem that BDNF levels are increased even by 14DIV for slice cultures (Figure 2A). What age of slice cultures were used in early papers (like Aoto, 2008; this information was missing from that paper)? This has been a contentious issue over the years, and since this offers a potential resolution to the discrepancies, more discussion of it is warranted.

RA dependent plasticity at the early age (14 DIV) occurs rapidly, within a few hours. Here the CNQX treatment was for 36 hours. Is this longer time necessary? Is RA induction slower at 21DIV? Or is only the frequency change slow and the amplitude change faster? Some explanation of this would be appropriate.

---

## [Author Response]

Essential revisions:This manuscript probes the mechanism of postsynaptic retinoic acid (RA) signaling on presynaptic function. BDNF has important roles in synaptic plasticity, but how retrograde BDNF signaling is controlled following synaptic inactivity is unclear. The authors use genetic tools to localize the action of different components of the pathway to pre- or post-synaptic compartments and use biochemical approaches to define a molecular link between retinoic acid and local translation of distinct BDNF transcripts. The findings presented here fill a gap in our knowledge regarding how presynaptic function is adaptively modulated by BDNF by highlighting the role of RA in this process. The experiments have been well-executed and the data provide compelling support for the model proposed by the authors. There are, however, some important issues to address:1) Evaluate the relative infection efficiency in CA3 neurons: This is critical in interpreting the negative result with RARa deltion. The pre-synaptic Cre expression would require a high percentage of CA3 neurons infected. From Figure 3, it appears this is achieved, but some information on this point would be helpful. Also in the results text for Figure 1, it should mention the use of AAV to express Cre (this is mentioned for Figure 3 but not Figure 1).

This reviewer is absolutely right that infection efficiency in CA3 is critical to the interpretation of our results regarding RARα, BDNF and TrkB’s involvement in homeostatic synaptic plasticity at the presynaptic site, especially when interpreting a negative result. Only slices infected with high titer AAV-Cre viruses and showed more than 90% infection efficacy in CA3 and no infection in CA1 were used for recordings (an example is shown in Figure 1 —figure supplement 1C). Additionally, for the data presented in the original submission, we used the same batch of AAV-Cre viruses (made in house) for the injection in all three types of cKO slices, thus the positive results in the presynaptic-TrkB KO slices served as quality control for our negative results in the presynaptic KO of RARα and BDNF.

We have added clear description of AAV-Cre injection scheme in the result text for Figure 1.

2) Define and discuss what the change in mEPSC frequency means: Changes in mEPSC frequency may reflect multiple synaptic adaptations – changes in presynaptic release probability or alterations in synapse numbers. The experiments done clearly delineate pre- and/or postsynaptic roles for RARα, BDNF, and TrkB, but alone, mEPSC frequency changes could reflect either enhanced release probability, growth of new synaptic contacts, or AMPAR recruitment to silent synapses. The authors should explicitly address these potential alternative interpretations early on in the paper. There is substantial evidence from other studies that the mEPSC frequency changes under these conditions reflect altered presynaptic release as the authors suggest, and making contact with this literature would make the conclusions of this paper that much more compelling.

This reviewer is correct in that a mEPSC frequency increase could mean multiple synaptic changes involved, with the three most prominent possibilities being new synapse formation, an increase in presynaptic release probability, and activation of silent synapses.

We ruled out the possibility of new synapse formation based on these two rationales: (1) the mEPSC frequency increase occurs only in older (> 21 DIV) but not young neurons in cultures. Synaptogenesis, on the other hand, is most prominent during the first two weeks of culture. Thus, if no new synapse formation is induced by RA at 14 DIV, it is unlikely that it is induced in older and more mature cultures (see also Aoto et al., Neuron 2008, where we showed in Figure S1 that RA treatment does not induce new spine formation). (2) Newly formed synapses tend to be smaller in size with smaller unitary mEPSC amplitudes and will bring down the average mEPSC amplitudes. We have consistently observed the mEPSC amplitude increases in older cultures at a magnitude comparable to those observed in younger cultures. Thus, we concluded that new synapse formation is unlikely a mechanism contributing to the increase in mEPSC frequency.

To examine possible changes in release probability, we performed paired-pulse ratio measurement of the evoked EPSCs at the CA3-CA1 synapses. We found that RA indeed decreased PPF. This is consistent with the results reported in Jakawich et al., 2010. We concluded that an increase in presynaptic release probability is likely a main mechanism driving increased mEPSC frequencies. This new data is included in Figure 1 —figure supplement 1A.

To examine possible silent synapse involvement, we also measured failure rate using minimal stimulation protocol before and after RA treatment in the older cultures. We found no significant decrease in failure rate as a result of RA treatment. This new data is included in Figure 1 —figure supplement 1B. Additionally, similar to our argument against new synapse formation, newly activated silent synapses tend to have smaller mEPSC amplitudes and their involvement would lead to less or no increase in mean mEPSC amplitudes averaged from all synapses. Instead, we have observed robust increase in both amplitude and frequency of mEPSCs. Lastly, using presynaptic TrkB KO, we show unequivocally the involvement of presynaptic signaling while turning on silent synapses typically requires postsynaptic signaling mechanisms. [1]

Taken together, we believe that the main mechanisms contributing to RA-induced mEPSC frequency increase is the enhanced release probability. We have added some discussions in the text to cover these points.

3) Discuss discrepancies with previous studies: The issue of age (of culture and animals) on whether this form of pre-synaptic plasticity occurs is important (although was also established in the earlier Wang paper). Some papers have reported frequency changes during HSP but many have reported no changes (including several from the Chen lab). Age/DIV seems to explain at least some of the difference, but I haven't gone through all the papers to check. It does seem that BDNF levels are increased even by 14DIV for slice cultures (Figure 2A). What age of slice cultures were used in early papers (like Aoto, 2008; this information was missing from that paper)? This has been a contentious issue over the years, and since this offers a potential resolution to the discrepancies, more discussion of it is warranted.

Our standard protocol for hippocampal slice culture uses P6-7 mouse pups for organotypic slice culture preparation, and DIV 7-12 slices for experiments. Our standard primary hippocampal cultures are made from P0 mouse or rat pups and experiments are usually conducted in DIV 12-14 neurons. This practice is widely adopted by many labs in the field, and was used by many of our early studies (and many other labs) in homeostatic plasticity because we were not aware of the age-dependent changes in presynaptic function at the time (Aoto et al., 2008; Turrigiano et al., 1998; Arendt et al., 2015; Kim and Ziff, 2014; to name a few). The mEPSC frequency increases were generally reported in studies using older primary cultures (DIV 17- 21). For example, in our hands and also in the studies by the Sutton lab, we observed frequency changes reliably in DIV21 culture (Jakawich et al., 2010; Wang et al., 2011). The Tsien lab has reported it in 17 DIV culture made from P2-P4 rat pups (Thiagarajan et al., 200). Note that depending on the exact culture protocol (serum-free Neurobasal media-based vs. serum-rich MEM-based media) and the age of pups used for culture preparation (E18, P0 or P4), the synaptic maturation course can be different in different studies at a given DIV time point. But overall, it is consistent across many studies that the frequency responses in synaptic homeostatic plasticity occurs in older and more mature cultures.

For this particular study, we had to ‘age’ the cultures in order to observe homeostatic presynaptic mEPSC increases. Specifically, DIV 21 primary cultured neurons made from P0-P1 mouse pups and DIV 21-25 organotypic slice culture made from P7-P8 mouse pups were used.

We have added more discussion on this part in the first paragraph of the result section.

4) Assess mIPSC changes: It is noted in the discussion that BDNF can increase mIPSC frequency. Does this happen here? During HSP, at least post-synaptically, excitatory and inhibitory synapses are inversely regulated. But it is possible miniature frequency is increased for both types. Or maybe not – either way, it would be interesting to define. Perhaps mIPSC amplitude goes down, but maybe that also hasn't been shown at these older ages/DIV. Looking at mIPSCs would extend impact and improve the paper.

The positive effect of BDNF on mIPSC reported in Bolton et al., 2000 was observed with treatment of BDNF in 3 DIV hippocampal cultures. At this age, no endogenous BDNF is available yet, so the effect likely reflects the developmental effect by exogenous BDNF in speeding up the GABAergic synapse development and maturation.

In response to reviewers’ inquiry, we examined RA’s effect on mIPSCs in 21 DIV hippocampal cultured slices, and found that both the amplitude and frequency of mIPSCs are significantly reduced by RA. Interestingly, we observed similar changes in mIPSCs in the post-critical period visual cortical circuit after visual deprivation or acute RA treatment (Zhong et al., J. Neurosci. 2018). This change is in the opposite direction to those observed for mEPSCs, and is consistent with the notion that synaptic silencing induces homeostatic increase of E/I ratio. This new data is included in Figure 1 —figure supplement 1D.

Next, to examine the effect of endogenous BDNF on basal inhibitory transmission and homeostatic plasticity, we measured mIPSCs in postsynaptic BDNF KO CA1 neurons in 23-25 DIV slice cultures. We observed a significant reduction in mIPSC amplitude in the cKO neurons. The mIPSC frequency showed a small trend of decrease (p = 0.06) in the cKO neurons. Treatment of RA did not further change the mIPSCs (please see Author response image 1). We believe this observation is consistent with the reported function of BDNF in inhibitory synapse development. Given the confounding developmental effect of BDNF on mIPSCs, we find it difficult to interpret the results of blocked mIPSC homeostatic plasticity in BDNF cKO neurons. Thus, we decided to leave the mIPSC data from the BDNF cKO neurons out of the current study and hope the reviewers will support our decision.

**Author response image 1. sa2fig1:** Effect of RA and postsynaptic BDNF deletion on mIPSCs. Quantification of mEPSC amplitudes (left) and frequencies (right) recorded from WT and postsynaptic BDNF KO neurons treated with DMSO or RA. **, p < 0.01; ***, p < 0.001, two-way ANOVA followed by Tukey’s test. Amp: F(1,117) = 17.55, p < 0.0001. Freq: F(1,117) = 15.59, p = 0.001. n/N = # of neurons/# of mice. Data shown as mean ± SEM.

5) Improved discussion and integration with previous work in the field: The results demonstrate a role for RA in controlling local dendritic BDNF synthesis and suggest RARα binding as a mechanism. As the authors note, previous work has similarly implicated the mTORC1 signaling pathway in local BDNF translation in response to synaptic inactivity. Parallel findings have been observed at the *Drosophila* neuromuscular junction. The authors may want to speculate in the discussion on whether these two pathways work together or in parallel during homeostatic plasticity.

We have included additional discussion on this as suggested.

6) Respond to several points of clarification:– Figures 1 and 3 include mEPSC traces where the background has been sharded to color coordinate with the respective histograms. This was likely done to make the figures easier to read, but the shading is somewhat distracting and really does the opposite. This is not a major issue as it does not affect the data or its interpretation, but the figures would be improved if either the traces themselves are colored or the shading is simply removed.

We have removed the background shades.

– The authors should consider superimposing the individual data points over the histograms for the electrophysiology data shown in Figure 1B-C and Figure 3B-F

We have added the scatter plots to the bar graph.

– The model shown in Figure 3 —figure supplement 1 very nicely summarizes the RA pathway, but could also be used to include other work in the field that implicates other pathways (e.g., the mTORC1 pathway).

We have modified this figure and added the mTORC1 pathway.

– In the introduction, the phrase 'runaway tendency' is vague and not obvious as to its meaning. Perhaps another phrase would be better.

We have changed the phrase to “self-reinforcing nature”. Hope it is clearer in its meaning.

– The figure labels could be improved. Pre-KO and post-KO can be misinterpreted –"Pre-KO" could be misinterpreted as being before the Cre expression manifested. Maybe "pre-syn KO" and "post-syn KO" would be easier to understand when initially looking at the figures?

We have changed the figure labels accordingly.

– The e-phys figures probably should show the data points (in the modern style) instead of just being bar graphs. And shouldn't the ephys statistics be 2-way ANOVAs instead of just t-tests? This likely won't impact the results but would be better.

Thanks for the suggestion. We have added the scatter plots to the bar graph. All ephys data are now analyzed with 2-way ANOVA and the results are reported in the legends.

References

1. Bergquist, S., D.K. Dickman, and G.W. Davis, A hierarchy of cell intrinsic and target-derived homeostatic signaling. Neuron, 2010. 66(2): p. 220-34.